# Replacement of SO_2_ with an Unripe Grape Extract and Chitosan during Oak Aging: Case Study of a Sangiovese Wine

**DOI:** 10.3390/antiox12020365

**Published:** 2023-02-03

**Authors:** Giovanna Fia, Silvio Menghini, Eleonora Mari, Cristina Proserpio, Ella Pagliarini, Lisa Granchi

**Affiliations:** 1Department of Agricultural, Food, Environmental, and Forestry Sciences and Technologies (DAGRI), University of Florence, via Donizetti, 6, 50144 Firenze, Italy; 2Sensory & Consumer Science Lab (SCS_Lab), Department of Food, Environmental and Nutritional Sciences (DeFENS), University of Milan, 20133 Milan, Italy

**Keywords:** natural antioxidant, sulfur dioxide, chitosan, wine, color, copigmentation, yeasts, bacteria, economic sustainability

## Abstract

The aim of this study was to evaluate the chemical, microbiological and sensory characteristics of a Sangiovese wine aged in barrique with the addition of an unripe grape extract (UGE) as an alternative to sulfur dioxide. Three samples were considered: control wine (TQ) with free SO_2_ of approximately 15 mg/L; sample A with chitosan (100 mg/L) and UGE (200 mg/L); and sample B with UGE (400 mg/L). The results achieved in this work demonstrated that the UGE, either alone or in combination with chitosan, was able to maintain the color characteristics of the Sangiovese wine and its sensory quality. Moreover, the addition of UGE contributed to an early and better stabilization of the color through the formation of polymeric pigments. The microbiological stabilization was comparable to SO_2_ when UGE was used at 200 mg/L in combination with chitosan. The market survey conducted in the present study confirmed how the use of UGE as an alternative to sulfitation was positively accepted by consumers, who are increasingly attentive not only to the quality of the wines they select but also to the sustainability of the production processes from which they derive and to the fact that they are not harmful to human health.

## 1. Introduction

Sulfur dioxide is an additive traditionally used to preserve wine from oxidation and microbiological spoilage. Sulfur dioxide prevents the enzymatic and chemical oxidation of the product and inhibits both lactic acid and acetic acid bacteria, some spoilage yeasts such as *Brettanomyces* spp. and other non-*Saccharomyces* species that can be found in grapes and wine [1,2]. The set of these properties contributes to maintaining the sensory quality of wine. Furthermore, sulfur dioxide is an inexpensive chemical additive. For these reasons, it is difficult to find efficient alternatives to sulfur dioxide. However, sulfur dioxide has been associated with several human health risks, including headaches, dermatitis, hives, abdominal pain, bronchoconstriction and anaphylaxis [3]. Over the past few years, consumer demand for wines free of potentially harmful compounds has increased, and several studies investigated the effectiveness of alternative additives [4]. The attention to wines without added sulfur dioxide has increased with the significant rise in the global demand for organic wines. In some markets, such as the US, the product that wants to be claimed as “100% organic” must always also be “sulfite free”; moreover, the characteristic “no added sulfites” is no longer an extra distinctive element but an indispensable feature for wines of higher quality segments.

As alternatives to sulfur dioxide, in order to achieve microbiological stability of the wine, the OIV (Organisation Internationale de la Vigne et du Vin) and the EU have approved various physical and chemical methods [5]. Indeed, some oenological products already available on the market have properties that can contribute to reducing the dose of sulfur dioxide added to wine. Oenological tannins are, for example, plant extracts from different botanical origins (oak gall, chestnut, quebracho, acacia, grape seeds and skins, tara, and mimosa) with interesting protecting properties against wine oxidation [6]. These compounds are extensively used due to the fact of their efficiency, mainly when they are added to the wine together with sulfur dioxide. However, the tannins doses normally used do not provide protection against pollutant microorganisms. Indeed, the phenolic compounds naturally occurring in grapes and wines, display various inhibiting or stimulating effects on the growth of lactic acid bacteria, depending on the polyphenols chemical structure and concentration, whereas the influences of phenolic compounds on yeasts growth have not been evaluated [5].

Recently, new phenolic extracts from various sources (black radish, almond peel and eucalyptus leaves) and from wine byproducts (vine shoots, skins and stems) have been proposed as potential substitutes of sulfur dioxide, with encouraging results [7,8,9,10,11,12]. The studies conducted with extracts obtained from the byproducts of the wine industry are particularly interesting, because they tend toward a model of regenerative production. However, few of these studies investigated the antimicrobial effect of the innovative extracts and their influence on wine sensory characteristics. 

In the wine industry, cluster thinning is a green pruning practice carried out to improve grapes’ quality at harvest. Thinning is often performed for high-quality wines according to the well-accepted idea that high crop yields delay ripening and reduce fruit and wine quality [13]. Unripe grapes (UGs) removed from the vine during thinning are usually abandoned in the field. Previous studies showed that UGs can be extracted and used as antioxidant and antibacterial additives or as fortifying agents in different types of food and beverages [14,15,16,17,18,19,20]. The properties of UGs are related to their richness of antioxidants, including phenols, stilbenes, glutathione, and vitamins. Organic acids are very concentrated in the UGs while sugars are low. In general, the concentration of these compounds depends on several factors, such as variety, vintage, climatic conditions, and stage of berry development [21]. 

At the market level, the use of UGE needs to be evaluated considering the growing demand for sustainable wines capable of safeguarding both the environment and consumer health. By reducing the inputs and waste occurring during wine production, the UGE offers an alternative that generates important process and product innovations, inspired by the principles of the circular economy [22]. The consumers could perceive wines with UGE addition as more sustainable [23], penetrating the niche of organic, biodynamic and natural wines leading them to be more willing to pay a higher price for such wines [24,25] as they perceive a better product both for the environment preservation and for their own health in terms of the so-called lifestyles of health and sustainability (LOHAS) [26].

In this work, an UGE was used as a stabilizing additive during the aging of a Sangiovese wine in comparison with sulfur dioxide. The chemical, microbiological and sensory characteristics of the wines were evaluated over twelve months of aging in barrique. Furthermore, an economic evaluation was also carried out for the entire UGE production and utilization activity, considering the costs that such innovation generates.

## 2. Materials and Methods

### 2.1. Unripe Grape Extracts

The extract was obtained from UGs (cv Sangiovese), following the method described in previous works [15,27]. The unripe grapes were handpicked, destemmed and crushed. Then, the grapes were extracted by maceration at 6 °C, for a period of 96 h, during which dry ice was added. The extract was racked and then decanted for 48 h, at 6 °C. Filtration was performed in order to remove large particles (i.e., 1 mm or more in diameter). The sugars were eliminated from the extract by ultrafiltration with the aid of a spiral wound configuration membrane with a molecular weight cut-off of 2500 Da (General Electric, Boston, MA, USA). After, the liquid extract was added with Arabic gum (2% *w*/*v*) (Nexira Food, Rouen Cedex, France) and then lyophilized. The dried extract was stored under vacuum in polyethylene pouches at room temperature and protected from the light.

### 2.2. Winemaking Process

The Sangiovese grapes were hand-harvested from the vineyards of Castello di Gabbiano (Treasury Wine Estate group) in 2019 and vinified at the cellar of the same farm using a stainless-steel vat with a capacity of 25 hL. After destemming and crushing, the must was added with 30 mg/L SO_2_. The must was inoculated with 20 g/hL of rehydrated active dried yeast (LALVEN ICV D254^TM^, Lallemand, Verona, 37060, Italy). Alcoholic fermentation and maceration were conducted at 30 °C for 7 days, and two pump-overs were performed daily. On the second day of fermentation, 40 g/hL of yeast nutrient (Nutriferm Ultra, Enartis) were added. At the end of fermentation (residual sugar less than 2 g/L), the pomace was pressed with a pneumatic press, and the press wine up to 1.8 bars was added back to the free run wine. The wine was transferred to a stainless-steel vat until the spontaneous malolactic fermentation (FML) was completed. At the end of the FML, the chemical parameters of the wine were measured: total acidity, 5.15 g/L (expressed as tartaric acid); pH 3.60; volatile acidity, 0.47 g/L (expressed as acetic acid); sugars, 0.10 g/L; alcohol degree, 14.55%; dry extract, 29.40 g/L; free sulfur dioxide, 8 mg/L; total sulfur dioxide, 20 mg/L; and lactic acid, 0.8 g/L. Then, the wine was racked and transferred in barrique for aging. All the barriques (Tonnellerie Baron, Les Gondes, France) used for the study were made with French oak. The barriques were two years old and medium toasted. Three samples were set-up in duplicate: TQ (15 mg/L free SO_2_, 140 mg/L Arabic gum); A (200 mg/L UGE, 100 mg/L chitosan); and B (400 mg/L UGE). After approximately 4 months of aging, the level of free sulfur dioxide of the TQ samples was checked and adjusted to approximately 15 mg/L, while 200 and 400 mg/L of UGE were added to samples A and B, respectively. Chemical, microbiological and sensory analyses were performed at the start (T0) and after 3 (T3), 6 (T6) and 12 (T12) months of aging in barrique.

### 2.3. Chemicals

All reagents and solvents were purchased from Sigma-Aldrich (Milan, Italy), except for methanol and ethanol, which were supplied by Carlo Erba (Milan, Italy). Potassium metabisulphite (Enartis), Chitosan Micro M (Enartis) and Oenogom instant (Laffort) were purchased from the market.

### 2.4. General Analysis

The total acidity, volatile acidity, pH, alcohol, total and free SO_2_, lactic acid and dry extract were evaluated according to the methods recommended by the International Organization of Vine and Wine (OIV) [28]. 

### 2.5. Color Intensity and Hue

The color intensity (CI) and hue (Hue) of the wine were measured using a Perkin Elmer Lambda 10 spectrophotometer (Waltham, MA, USA), following the method described by other authors [29]. The absorbance (A) at 420, 520 and 620 nm was measured, and the CI and Hue were calculated as follows: CI = (A420 +A520 + A620) × 10 and H = A420/A520.

### 2.6. CIELab Chromatic Characteristics

The CIELab space was used to evaluate the color of the red wine. The following parameters were calculated: lightness from black to white (L*), color from blue to yellow (b*), color from red to green (a*), chroma or saturation (C*) and hue angle (H*), following the method OIV-MA-AS2-11: determination of chromatic characteristics according to CIELab [30].

### 2.7. Copigmentation

The copigmentation was evaluated according to the method proposed by Boulton et al. [31]. The following parameters were considered: color due to the fact of copigmented anthocyanins (AC); color fraction due to the fact of copigmentation (COP); color due to the presence of total anthocyanins (TAs); color fraction due to the presence of free anthocyanins (AL); color due to the fact of polymeric pigments (Eps); color fraction due to the fact of polymeric pigments (PPs); flavanol cofactor index (FC); and total phenol index (TPI280).

### 2.8. Total Phenols

The total phenols (TPs) were quantified according to the Folin–Ciocalteu method [32]. The phenolic compounds were removed from 1 mL of wine and a 10% UGE solution with a C18 Sep-pak cartridge (Waters, Milan, Italy) [33]. A volume of 4 mL of sodium carbonate (10%, *w*/*v*) was added to 1 mL of sample and left to stand for 5 min. A volume of 1 mL of diluted Folin–Ciocalteu reagent was added to the mixture. Then, the samples were left in the dark for 90 min at room temperature. After, the absorbance (700 nm) was measured with a Perkin Elmer Lambda 10 spectrophotometer (Waltham, MA, USA). Solutions of (+)-catechin ranging from 5 to 500 mg/L were used as references. The TPs content of the samples was expressed as mg of (+)-catechin equivalents (mg CAT eq)/g of UGE or (mg CAT eq)/L of wine.

### 2.9. DPPH 

The antioxidant activity (AA) was determined by a 2,2-diphenyl-1-picrylhydrazyl (DPPH) method [34]. For the reaction, 0.1 mL of the sample was mixed with 3.9 mL of the DPPH solution (6 × 10^−5^ M), prepared in methanol. A volume of 0.1 mL of the sample was added to 3.9 mL of methanol, as the reference. To obtain the maximum absorbance of DPPH, 0.1 mL of methanol was mixed with 3.9 mL of a 6 × 10^−5^ M DPPH solution. After 30 min at 30 °C in the dark, the decrease in the absorbance at 515 nm of the reaction mixture was measured against the methanol reference sample. Trolox standard solutions at concentrations ranging from 10 to 600 μmol/L were prepared in absolute ethanol. The antioxidant activity is expressed as μmol of Trolox/L of wine.

### 2.10. FRAP

The antioxidant activity was measured using the FRAP assay [35]. Trolox solutions, in the range of 0.05–1.18 mM, were prepared and assayed for the calibration curve. The FRAP reagent (acetate buffer: 300 mM:2,4,6-tris(2-pyridyl)-s-triazine 9.99 mM:FeCl3·6H2O 20 mM) (10:1:1)) was prepared. For the sample, 150 μL wine sample, diluted at 1:15 or 1:25 with methanol, or 150 μL of the Trolox standard solutions were mixed with 2.85 mL of FRAP reagent. All the mixtures were left for 30 min. Then, the absorbance was measured spectrophotometrically at 595 nm. The antioxidant activity is expressed as μmol of Trolox/L of wine.

### 2.11. ABTS

The antioxidant activity was measured by the ABTS method [36]. A 7 mM 2,2’-azinobis 3-ethylbenzothiazoline-6-sulphonic acid (ABTS) solution with the addition of potassium persulfate (2.45 mM) was prepared. The solution was left in the dark for 16 h. A calibration curve, ranging from 0.05 to 2.11 mM, was prepared from a 5 mM stock solution of Trolox. For sample, the wine, previously diluted at 1:7 or 1:15 with methanol, and 30 μL of the Trolox standard solutions were mixed with 2.97 mL of a ABTS•+ radical cation solution. After 30 min in the dark, the absorbance of the reaction mixture was spectrophotometrically measured at 734 nm. The antioxidant activity is expressed as μmol of Trolox/L of wine.

### 2.12. Microbiological Analysis

The quantification of yeasts, lactic acid bacteria (LAB), and acetic acid bacteria (AAB) was carried out by plate counts and selective culture media. The yeasts were enumerated on WL nutrient agar (Oxoid) [37], containing sodium propionate (2 g/L) and streptomycin (0.3 g/L). Lactic acid bacteria were quantified on MRS ISO agar (Oxoid) [38], with the addition of fructose (5 g/L), cysteine (0.5 g/L), tomato juice broth (2.5 g/L), agar (6 g/L) and pimaricin (0.05 g/L). The acetic bacteria were quantified on LF Agar Medium (glucose, 10 g/L; yeast extract, 5 g/L; peptone, 5 g/L; tomato juice broth, 2 g/L) with the addition of pimaricin (0.05 g/L) and penicillin (0.025 g/L). To enumerate *B. bruxellensis*, a representative number of colonies showing a green color, round morphology, and yellow halo were picked up and analyzed using RFLP-PCR of rITS for their identification [39].

### 2.13. Sensory Evaluation

A total of 30 subjects (56% women; aged between 22 and 50 years; mean age: 32.1 ± 7.8 years) were recruited among students and employees of the Faculty of Agriculture and Food Sciences of the University of Milan (Italy). Only subjects who like and regularly consume wine and were free of food intolerances and allergies were involved. The study was conducted in accordance with the Declaration of Helsinki and approved by the Ethics Committee of the University of Milan (protocol code: 30/21; date of approval: 18 March 2021). Informed consent was obtained from all subjects involved in the study. 

The wine samples were subjected to a triangle method [40]. The participants attended four sessions (T0, T3 T6 and T12) at the Sensory and Consumer Science Laboratory (SCS_Lab) of the Department of Food, Environmental and Nutritional Sciences of the University of Milan, designed according to ISO guidelines [41]. The evaluations were conducted between 11:30 am and 13:30 pm, and the judges were asked to consume only water and not smoke during the 2 h before the evaluation. 

During the evaluations, the following comparisons were made: TQ vs. A and TQ vs. B at the four times of aging (T0, T3, T6 and T12). The judges were presented with three samples simultaneously (two equal) and were asked to identify within each triad which sample was perceived as different considering all the sensory characteristics related to sight, smell, taste, flavor and body. To minimize adaptation, a 3 min break occurred between triads and panelists were instructed to take additional breaks when they desired. The samples were swallowed and retasting was permitted. The evaluation was carried out in a single run, and the presentation sequences of the samples were randomized judge by judge. The samples were stored at room temperature away from light and heat sources and served in glass goblets encoded with three-digit numbers. Each glass was fitted with a top cap to avoid the dispersion of volatile compounds. The presentation order of the treatment comparisons was counterbalanced across panelists, and the sample presentation was randomized within triads.

Instructions were provided to the judges regarding the evaluation procedure in both written and verbal formats. They were instructed to focus on all sensory characteristics evaluating the samples one at a time, keep the samples capped when not being tasted, proceed at their own pace, and to cleanse the palate with natural mineral water and crackers to desaturate the sensory receptors between samples. 

The session took approximately 30 min. The data acquisition was performed with Fizz v2.31 software (Biosystèmes, Couternon, France).

### 2.14. Economic Evaluation

An economic analysis was developed to evaluate how close to market is the use of the UGE. This economic analysis was carried out both at the company and market levels. The company analysis concerned the effects that the process innovation introduced had in terms of organization and the internal management of production costs. The cost analysis was carried out with the analytic accounting methodology of full costing, considering all expenses (fixed–variable, direct–indirect and explicit–figurative) sustained in the production and use of UGE. With regard to the market analysis, a direct survey was conducted on a significant number (1000) of wine consumers in Italy. The direct survey, conducted over the period 2020–2021, started from the analysis of consumer behavior compliant with sustainable lifestyles in terms of LOHAS: subjects with a high level of LOHAS were recruited as targets for wines without added sulfites, testing their willingness to pay for such a product.

### 2.15. Statistical Analysis

All analyses were conducted in triplicate. The statistical analysis was carried out using the XLSTAT statistical and data analysis solution (2021) (Addinsoft, Paris, France). The analysis of variance (ANOVA) (least significant differences (LSD): 5% level) and the Tukey’s least significant difference (LSD) test were conducted for each chemical and microbiological variable in order to assess the significant differences among wines (*p*-value < 0.05). For the sensory evaluation, the number of correct responses (identification of the different sample in the triad) was counted, and the significance was determined using a minimum number of correct responses (ISO 4120:2021). If the number of correct responses is greater than or equal to the tabulated number, a perceptible (*p*-value < 0.05) difference exists between the samples.

## 3. Results and Discussion

### 3.1. Chemical Analyses

The phenolic concentration of UGE was 20.4 mg CAT eq/g of powder and the antioxidant activity, evaluated using the DPPH method, was 33.8 μmol Trolox/g. In this work, the doses of phenols added to the wine were chosen on the basis of the results of previous studies in which the antioxidant effect of the phenolic extracts obtained from the viticultural byproducts was evaluated on the must and wine [9,16]. The UGE was added to the wine at doses of 200 and 400 mg/L providing approximately 4 (sample A) and 8 mg (sample B) of phenols/L of wine, respectively. Previous studies showed that UGEs contained high levels of organic acids, and their composition may include glutathione and water-soluble vitamins, which could contribute to the antioxidant activity of the extracts [15,16]. The same authors tested the UGEs as an alternative to sulfur dioxide to protect the color of red wine from oxidation during aging in steel tanks, with promising results [21]. However, microbiological wine spoilage during oak aging is particularly feared. For this reason, chitosan (100 mg/L) was added as a microbiological stabilizer in sample A, which received the lowest dose of UGE.

Table 1 shows the initial composition (T0) of each wine treated with different stabilizing agents. At the start (T0), immediately after the addition of SO_2_, UGE and chitosan, some color parameters of the samples showed small but significant differences. In particular, samples A and B showed values of Abs520, Abs520% and CI that were significantly higher than that observed in the TQ. In addition, the CIELab analysis showed that samples A and B had the following characteristics: (i) the lowest L* values, related to the darkness of the color; (ii) the highest level of a*, related to the red shades of the color; (iii) the highest C* values, related to the saturation of the color. Hence, these results indicate that sample TQ was less colored, saturated, and lighter than the other samples. These results are compatible with the bleaching effect of sulphur dioxide on the anthocyanins of red wine [42,43]. Regarding the copigmentation parameters, the samples did not show significant differences except for the cofactor content (FC) that was the lowest in sample TQ. Previous results showed that extracts from unripe grapes are rich in flavonoids [14,15]. In the A and B samples, the addition of UGE could have contributed to cofactor content in terms of flavonoids which are cofactors in the copigmentation reaction [31]. At the start, the total phenol content (TP) evaluated by the Folin–Ciocalteu method was not significantly different among the samples studied (Table 1). The antioxidant activity of the different samples was measured by the DPPH, FRAP and ABTS methods (Table 1). 

In agreement with the results obtained by other authors, the highest values of antioxidant activity were found when the ABTS method was used, followed by FRAP and DPPH [9]. However, when the samples were studied with the same method, there were no significant differences in the antioxidant activity among the samples studied.

Table 2 shows the final composition (after 12 months) of the different wine samples that were studied. After twelve months of aging, the wine samples showed differences in some color and copigmentation parameters, while the total phenol content and antioxidant activity were similar (Table 2). In particular, the TQ showed the lowest Abs520 and CI values and the highest L* value, indicating that TQ was slightly less colored and lighter than the other samples. According to the present results, other authors who analyzed the color parameters of red wines elaborated with stem or shoot extracts as alternative antioxidants to SO_2_ found that wines treated with SO_2_ had significantly higher luminosity values (L*) and lower color intensity than the wines elaborated with the alternative antioxidants [44,45]. The hue of color, b* and H* of the samples were similar at this stage of aging, highlighting that the wine color reached a similar level of oxidation.

A copigmentation analysis provides information concerning the evolution of wine color. Copigmentation is a complex phenomenon influenced by the composition of wine in term of anthocyanins, cofactors, metals, ethanol, and pH [46,47,48]. After twelve months (T12), the samples with the UGS showed significant differences for almost all copigmentation parameters when compared with sample TQ. Sample TQ showed the highest values of color due to the copigmented anthocyanins (C), color fraction due to the copigmentation (COP) and color fraction due to the fact of free anthocyanins (AL) and the lowest values of color due to the fact of polymeric pigment (Ep) and color fraction due to the fact of polymeric pigment (PP) and cofactor content (FC). There were some small but significant differences between sample A and sample B. In particular, sample B, having received the highest dose of UGE, showed higher AC and COP and lower Ep and PP values than sample A. A similar effect of the natural antioxidant extracts used in comparison with SO_2_ was highlighted by other authors [45]. These authors found that the red wine treated with stem or shoot extracts had significantly higher polymerization percentage values than the wine treated with SO_2_. The formation of polymeric pigments, normally accompanied by a decrease in free anthocyanins, contributes to the stabilization of red wine color [6,31].

The evolution of total phenols and antioxidant activity during aging is showed in Figure 1. Over the course of twelve months very few significant differences in the antioxidant activity of the samples were detected. In particular, it is possible to note that after three months, samples A and B had a significantly lower concentration of total phenols in comparison to that of sample TQ. Moreover, at the start (T0) and after three months, the values of ABTS were significantly lower in sample B. 

Accordingly, Salaha et al. [12] did not find significant differences in the antioxidant activity between wines of different varieties treated with normal doses of SO_2_ and the same wines treated with reduced doses of SO_2_ combined with black radish extracts (Rafhanus niger) and ascorbic acid. 

The evolution of the TP (Figure 1a) of all the wine samples showed a positive trend, mainly from the third month. In general, the TP content of the wines increased over aging due to the fact of contact with toasting oak [44]. The antioxidant activity evaluated by DPPH, FRAP and ABTS (Figure 1b–d), in general, decreased from the start to the end of the aging period. The highest decrease was observed in the antioxidant activity assayed by the DPPH method. The antioxidant properties of the wines, measured by the DPPH, FRAP and ABTS methods, during the aging period (Figure 1) indicated that the addition of UGE (400 mg/L) or UGE (200 mg/L) and chitosan (100 mg/L) can provide a level of protection from oxidation similar to that exerted by SO_2_. It is well documented that the antioxidant activity of phenol compounds can be related to different properties, including the capacity of scavenging superoxide radicals, consuming dissolved oxygen, reducing ferric iron (Fe^3+^) to ferrous iron (Fe^2+^) and chelating Fe^2+^ (6).

The evolution of some parameters of the wine treated with different stabilizing products during the aging period is shown in Figure 2. The sample TQ had the significantly lowest CI and highest L* values at all stage of aging (Figure 2a,c). These results were a consequence consistent with the further addition of SO_2_ made at the fourth month of aging, and its effect of bleaching on anthocyanins [42,43]. The differences in the AL, COP and PP content among the samples were detectable as early as the third month, and they were maintained until the end of aging (Figure 2). The TQ samples had the significantly highest AL and lowest PP values from T3 to T12 (Figure 1f,h). Moreover, the COP values of sample TQ were significantly the highest from the third month to the end of aging (Figure 1g). During this period, the rate of stable polymeric pigments (PP) formation in samples A and B was higher than that of TQ and was accompanied by a decrease in free anthocyanins (AL). These results highlighted that the addition of UGE could contribute to a faster stabilization of color. Hence, at the same stage of evolution, the wine aging with the addition of UGE contained more stable pigments towards oxidation phenomena in comparison to those of wine sample (TQ) aging with SO_2_ [31,47].

### 3.2. Microbiological Analysis

Microbial populations occurring in Sangiovese wines at the beginning and after 3, 6 and 12 months of aging are shown in Table 3. At the start, after the accomplishment of the malolactic fermentation, the concentrations of yeasts and acetic acid bacteria (AAB) were below the limit of detection, while lactic acid bacteria (LAB), belonging to the *Oenococcus oeni* species, attained concentrations between 10^4^ and 10^5^ UFC/mL that are values usually found at the end of malolactic fermentation [2]. During the aging period, in the control wine (TQ) treated with sulphur dioxide as well as in the A and B wines containing UGE at different concentrations, LAB decreased to cell densities below the limit of detection except in the sample B after 12 months of aging in which they attained at 10^4^ UFC/mL. On the contrary, after 3, 6 and 12 months of aging in barriques, the yeast populations increased both in the TQ wine and in the treated wines. In particular, after 3 months (T3) the concentration of yeasts in the TQ wine was significantly higher than in the A and B wines. However, in the TQ and A samples, the yeast population consisted exclusively of the *Saccharomyces cerevisiae* species, whereas in the B sample in which UGE was added at the highest concentration (400 mg/L) but not chitosan, *Brettanomyces bruxellensis* occurred, although at a very low concentration, corresponding, on average, to 36 CFU/mL. After 6 (T6) and 12 months (T12) of aging, the yeast concentrations showed no significant differences in the control wine and in two treated wines. The presence of *B. bruxellensis* was confirmed only in the B wine, where it reached a maximum cell density of 125 and 25 CFU/mL after 6 and 12 months of aging, respectively. Although these values were below the threshold of 10^3^ UFC/mL that is considered critical for the production of ethyl phenols [49], they could represent a potential cause of spoilage. On the contrary, in the A sample containing UGE and chitosan, as well as in the TQ sample containing sulfur dioxide, *B. bruxellensis* was not detected. Hence, according to these findings, the addition of UGE at 400 mg/L contributed to the microbiological stability of Sangiovese wine by controlling the bacteria population growth until the sixth month of aging, similar to 15 mg/L sulphur dioxide. For a longer aging time, the use of UGE at 200 mg/L in combination with chitosan demonstrated its effectiveness in controlling both *B. bruxellensis* and bacteria populations.

### 3.3. Sensory Evaluations

The results of the sensory evaluations are shown in Table 4. As can be seen, at T0 no significant differences were observed between the pairs of samples that were considered. Therefore, both samples containing an increasing concentration of unripe grapes extract (A and B) were not perceived as different by the consumers compared to the wine sample containing sulfur dioxide (TQ). During aging, after three, six and twelve months, the differences between the wines were still not significant. This means that the panel of judges perceived these products as similar in terms of the sensory characteristics related to sight, smell, taste, flavor and body, suggesting a promising role in the substitution of sulfur dioxide in the winemaking process. On the contrary, previous results revealed that the use of alternatives to SO_2_ are responsible for important changes in sensory attributes. Among the possible natural alternatives, it has been recently suggested that natural extracts obtained from grape seeds and American oak wood by accelerated extraction with subcritical water did not cause sensory defects in wines and led to a greater aromatic complexity and were positively evaluated by consumers [45]. It has also been depicted that hydroxytyrosol, as an alternative to sulfur dioxide in Syrah red wines, led to significant differences in sensory analysis [7]. Indeed, the hydroxytyrosol addition affected the aroma intensity (black fruit and mature fruit) but without leading to any aromatic defects (such as yeast, chemical and rancid aromas). Accordingly, some authors described a greater intensity of the olfactory descriptor in red wines elaborated with a colloidal silver complex instead of SO_2_ [50]. Similarly, Santos et al. [51] demonstrated that high-pressure treatments as an alternative to SO_2_ of red wines significantly altered aroma and taste perception. 

### 3.4. Economic Evaluation

With regard to the economic evaluations, the results obtained at the farm level show that the extract can be easily implemented in a winery without any particular initial investment, relying on resources (i.e., capital and labor) normally available in any cellar. The exception is the drying process, for which external services must be used. 

The full cost analysis carried out for the case study examined in this paper revealed that the farm, in order to produce the necessary UGE by itself, has to incur a total cost ranging from 0.79 to 1.18 per gram of obtained dried antioxidant extract. As explained in the Section 2.14, this cost was calculated by examining the entire process of producing the extract, starting from harvesting the unripe grapes to obtaining the final product; the process inputs (i.e., capital and labor) were considered for each operation, and the direct and indirect costs to be incurred by the entrepreneur for their use were quantified. The total cost range was mainly due to the dry extract yield, which can range from 20 to 30 g per liter of processed liquid extract.

The full cost analysis carried out for the case study revealed a total cost for the production of UGE ranging from EUR 0.79 to 1.18 per gram of dried antioxidant extract; the range was mainly due to the dry extract yield, which can range from 20 to 30 g per liter of processed liquid extract.

Given that 400 and 800 mg/L UGE were used for samples A and B, respectively, the cost of producing and adding UGE ranges from EUR 0.31–0.47 per liter of wine for sample A and EUR 0.63-0.94 per liter for sample B (Table 5).

These additional costs, which are minimal in absolute terms, are even smaller considering that the innovation examined in this study is mostly intended for medium–high quality wines, targeting consumers who select the product with a good willingness to pay, preferring sustainable and healthy wines. Regarding these last two aspects, the market survey carried out in the present study showed that more than 89% of respondents had a lifestyle particularly focused on personal health and sustainability (LOHAS); a wine without added sulfites, perceived as “healthier” and environmentally friendly, is the perfect synthesis to respond to these market trends and satisfy this broad target.

Respondents expressed a clear willingness to pay a premium price for a wine without added sulfites, especially if it is also organic: the direct survey reveals that this premium price is 8% for conventional NSA wines, while this value rises to over 13% if the wine is also organic.

## 4. Conclusions

The results of the present study suggest that UGE could be applied as promising alternatives to sulfur dioxide in the winemaking processes. Indeed, UGE either alone or in combination with chitosan was able to maintain the color characteristics and contributed to an early and better stabilization of the color through the formation of polymeric pigments of the Sangiovese wine. No significant differences in the antioxidant activity after 3, 6 and 12 months of aging were found, and the samples added with different UGE concentrations were perceived as comparable to the wine sample containing sulfur dioxide (TQ) in terms of sensory quality. The microbiological stabilization was comparable to SO_2_ when UGE was used at 200 mg/L in combination with chitosan. The market survey conducted in the present study confirmed how the use of UGE as an alternative to sulfitation was positively accepted by consumers, who are increasingly attentive not only to the quality of the wines they select but also to the sustainability of the production processes from which they derive and to the fact that they are not harmful to personal health. The substitution of SO_2_ by UGE should be carried out in future studies taking into consideration the vintage effect in wines.

## Figures and Tables

**Figure 1 antioxidants-12-00365-f001:**
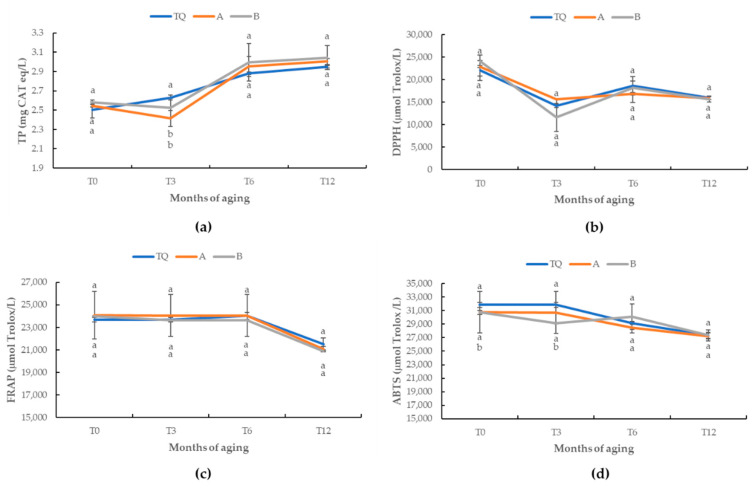
Evolution of antioxidant activity during the aging period (12 months) in the wine treated with different stabilizing products: (**a**) total phenols (TP); (**b**) DPPH values; (**c**) FRAP values; (**d**) ABTS values. Treatments: TQ (SO2); A (UGE, 200 mg/L and chitosan); B (UGE, 400 mg/L). Different letters indicate significant differences among the samples according to the Tukey’s least significative difference (LSD) test.

**Figure 2 antioxidants-12-00365-f002:**
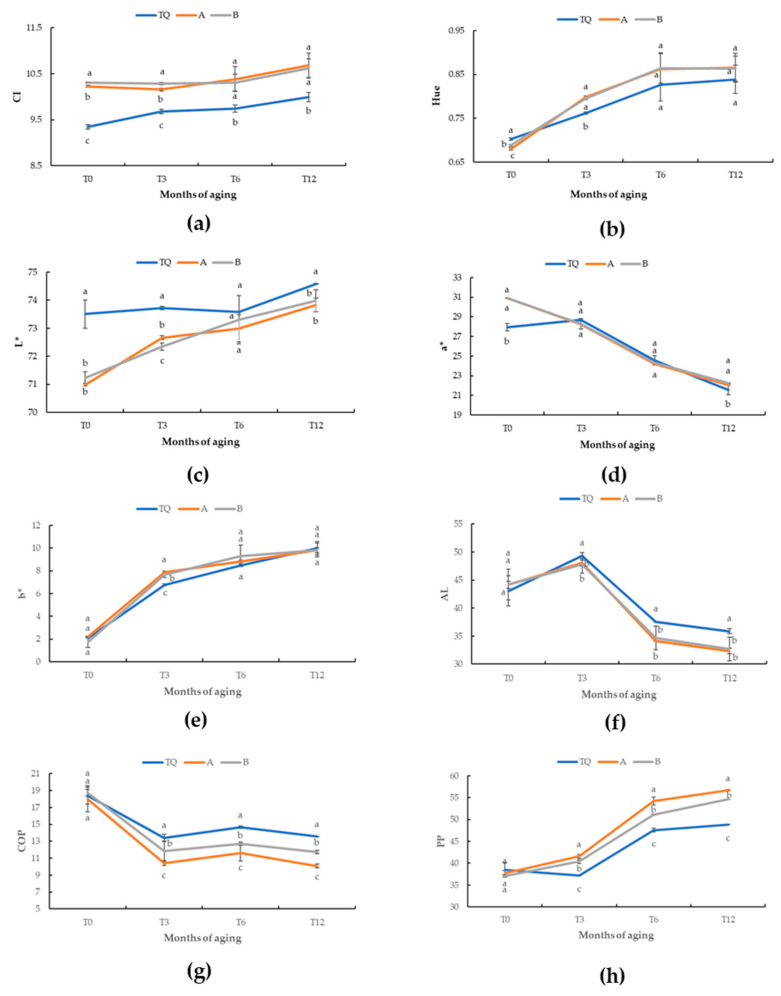
Evolution of several parameters during the aging period (12 months) in the wine treated with different stabilizing products: (**a**) color intensity (CI); (**b**) color hue (Hue); (**c**) lightness (L*); (**d**) red/green (a*); (**e**) blue/yellow (b*); (**f**) color fraction due to the fact of free anthocyanins (AL); (**g**) color fraction due to the fact of copigmentation (COP); (**h**) color fraction due to the fact of polymeric pigments (PP). Treatments: TQ (SO_2_); A (UGE, 200 mg/L and chitosan); B (UGE, 400 mg/L). Different letters indicate significant differences among the samples according to the Tukey’s least significative difference (LSD) test.

**Table 1 antioxidants-12-00365-t001:** Chemical parameters of the wine samples at the start (T0) after the addition of different stabilizing products.

	TQ	A	B	F-Value
Abs420	3.43 ± 0.01 c	3.68 ± 0.01 b	3.73 ± 0.02 a	749.65 ***
Abs520	4.88 ± 0.02 b	5.42 ± 0.01 a	5.43 ± 0.01 a	2067.66 ***
Abs620	1.03 ± 0.01 c	1.12 ± 0.01 b	1.14 ± 0.00 a	810.00 ***
ABS420%	36.70 ± 0.08 a	36.03 ± 0.07 c	36.22 ± 0.09 b	98.73 ***
Abs520%	52.26 ± 0.08 c	52.99 ± 0.10 a	52.66 ± 0.08 b	93.22***
Abs620%	11.02 ± 0.04 b	10.98 ± 0.03 b	11.11 ± 0.01 a	27.07 ***
CI	9.34 ± 0.03 c	10.23 ± 0.02 b	10.31 ± 0.03 a	1846.91 ***
Hue	0.702 ± 0.003 a	0.680 ± 0.003 c	0.682 ± 0.003 b	98.90 ***
L*	73.50 ± 0.38 a	70.99 ± 0.11 b	71.23 ± 0.21 b	164.18 ***
a*	27.92 ± 0.28 b	30.92 ± 0.15 a	30.90 ± 0.17 a	388.31 ***
b*	2.09 ± 0.09 a	2.23 ± 0.15 a	2.05 ± 0.11 a	3.27 (*)
C*	28.01 ± 0.28 b	31.02 ± 0.14 a	30.97 ± 0.16 a	404.00 ***
H*	13.31 ± 0.63 a	13.89 ± 1.08 a	13.77 ± 0.23 a	1.07
AC	0.79 ± 0.03 a	0.79 ± 0.06 a	0.83 ± 0.02 a	2.11
COP	18.43 ± 0.78 a	18.03 ± 1.19 a	18.73 ± 0.33 a	1.04
TA	1.85 ± 0.15 a	1.93 ± 0.01 a	1.97 ± 0.01 a	2.46
AL	43.06 ± 3.65 a	44.19 ± 0.74 a	44.18 ± 0.48 a	0.54
Ep	1.66 ± 0.14 a	1.651 ± 0.004 a	1.65 ± 0.01 a	0.02
PP	38.50 ± 3.32 a	37.77 ± 0.45 a	37.08 ± 0.20 a	0.80
FC	6.84 ± 0.11 b	6.93 ± 0.05 ab	7.00 ± 0.03 a	6.77 **
TPI280	54.50 ± 0.21 a	54.22 ± 0.11 b	53.99 ± 0.13 b	14.29 ***
TP	2.50 ± 0.13 a	2.55 ± 0.06 a	2.58 ± 0.07 a	0.95
DPPH	22,052 ± 2211 a	22,891 ± 1221 a	24,076 ± 1400 a	0.68
FRAP	23,677 ± 178 a	24,091 ± 2102 a	24,009 ± 2291 a	0.06
ABTS	31,842 ± 386 a	30,770 ± 3048 a	30734 ± 270 a	0.20

All parameters are given as the mean ± standard deviation. Treatments: TQ (SO2); A (UGE, 200 mg/L and chitosan); B (UGE, 400 mg/L). The different letters indicate significant differences among the samples according to the Tukey’s least significative difference (LSD) test. Significant values are shown according to the following: (*) *p*-value < 0.1; ** *p*-value < 0.01; *** *p*-value < 0.001.

**Table 2 antioxidants-12-00365-t002:** Chemical parameters of the wine samples aged for twelve months (T12) in the presence of different stabilizing products.

	TQ	A	B	F-Value
Abs420	4.00 ± 0.09 b	4.32 ± 0.12 a	4.30 ± 0.11 a	11.60 **
Abs520	4.77 ± 0.08 b	4.99 ± 0.02 a	4.95 ± 0.05 a	16.81 ***
Abs620	1.20 ± 0.01 b	1.37 ± 0.05 a	1.34 ± 0.03 a	25.37 ***
ABS420%	40.09 ± 0.71 a	40.42 ± 0.41 a	40.58 ± 0.53 a	0.75
Abs520%	47.83 ± 0.81 a	46.71 ± 0.70 a	46.75 ± 0.70 a	3.10 (*)
Abs620%	12.07 ± 0.12 b	12.85 ± 0.30 a	12.66 ± 0.16 a	18.69 ***
CI	9.99 ± 0.11 b	10.69 ± 0.21 a	10.60 ± 0.16 a	28.84 ***
Hue	0.83 ± 0.03 a	0.86 ± 0.02 a	0.86 ± 0.02 a	1.75
L*	74.58 ± 0.02 a	73.75 ± 0.41 b	74.05 ± 0.47 ab	6.19 *
a*	21.55 ± 0.37 b	22.03 ± 0.05 a	22.22 ± 0.04 a	44.35 ***
b*	9.98 ± 0.47 a	9.80 ± 1.35 a	9.77 ± 0.61 a	0.09
C*	23.66 ± 0.15 b	24.40 ± 0.52 a	24.44 ± 0.29 a	20.96 ***
H*	2.11 ± 0.15 a	2.04 ± 0.33 a	2.03 ± 0.14 a	0.21
AC	0.63 ± 0.01 a	0.33 ± 0.03 c	0.47 ± 0.07 b	66.86 ***
COP	13.57 ± 0.11 a	10.01 ± 0.24 c	11.62 ± 0.32 b	251.60 ***
TA	1.54 ± 0.10 a	1.41 ± 0.09 a	1.42 ± 0.12 a	2.16
AL	35.88 ± 0.48 a	32.46 ± 0.33 b	32.72 ± 1.63 b	19.97 ***
Ep	2.265 ± 0.002 c	2.57 ± 0.01 a	2.528 ± 0.004 b	3383.12 ***
PP	48.88 ± 0.07 c	56.68 ± 0.35 a	54.70 ± 0.41 b	954.49 ***
FC	7.44 ± 0.04 b	7.93 ± 0.03 a	7.91 ± 0.03 a	309.03 ***
TPI280	59.11 ± 0.46 b	60.58 ± 0.66 a	60.37 ± 0.45 a	14.33 ***
TP	2.88 ± 0.09 a	2.89 ± 0.13 a	2.99 ± 0.15 a	1.17
DPPH	15,931 ± 312 a	15,812 ± 606 a	15,812 ± 606 a	1.54
FRAP	21,514 ± 534 a	21,050 ± 243 a	20,902 ± 145 a	0.04
ABTS	27,343 ± 457 a	27,255 ± 431 a	27,340 ± 843 a	1.00

All parameters are given as the mean ± standard deviation. Treatments: TQ (SO2); A (UGS, 200 mg/L and chitosan); B (UGS, 400 mg/L). Different letters indicate significant differences among the samples according to the Tukey’s least significative difference (LSD) test. Significant values are shown according to the following: (*) *p*-value < 0.1; * *p*-value < 0.05; ** *p*-value < 0.01; *** *p*-value < 0.001.

**Table 3 antioxidants-12-00365-t003:** Yeasts, lactic acid bacteria (LAB) and acetic acid bacteria (AAB) occurring in wines at 0, 3, 6 and 12 months of aging. Treatments: TQ (SO_2_); A (UGE, 200 mg/L and chitosan); B (UGE, 400 mg/L). All parameters are given as the mean ± standard deviation. Different letters indicate significant differences among the samples (ANOVA; *p* < 0.05).

Aging (Months)	Sample	Yeasts (UFC/mL)		LAB (UFC/mL)		AAB (UFC/mL)
	TQ	<20		(2.06 ± 0.65) × 10^5^	a	<10
T0	A	<20		(2.16 ± 0.55) × 10^4^	b	<10
	B	<20		(8.06 ± 2.58) × 10^4^	c	<10
	TQ	(3.00 ± 0.56) × 10^2^	a	<10		<10
T3	A	(1.00 ± 0.50) × 10	b	<10		<10
	B	(5.33 ± 0.58) × 10 ^§^	c	<10		<10
	TQ	(1.30 ± 0.28) × 10^2^	a	<10		<10
T6	A	(2.45 ± 0.77) × 10^2^	a	<10		<10
	B	(7.75 ± 4.86) × 10^2 †^	a	<10		<10
	TQ	(6.39 ± 9.70) × 10^3^	a	<10		<10
T12	A	(6.35 ± 5.66) × 10^3^	a	<10		<10
	B	(6.24 ± 4.23) × 10^3 ‡^	a	(1.56 ± 0.40) × 10^4^		<10

^§^ 67% Brettanomyces bruxellensis; ^†^ 16% Brettanomyces bruxellensis; ^‡^ 0.4% Brettanomyces bruxellensis. Different letters indicate significant differences among the samples according to the Tukey’s least significative difference (LSD) test.

**Table 4 antioxidants-12-00365-t004:** Triangle method results for the pairs of Sangiovese wine samples analyzed at 0, 3, 6 and 12 months of aging.

Aging (Months)	Sample	Correct Responses	*p*-Value
T0	TQL–AL	11	0.41 n.s.
	TQL–BL	8	0.83 n.s.
T3	TQL–AL	12	0.20 n.s
	TQL–BL	8	0.75 n.s.
T6	TQL–AL	14	0.09 n.s.
	TQL–BL	8	0.83 n.s.
T12	TQL–AL	9	0.98 n.s.
	TQL–BL	5	0.71 n.s.

n.s., not significant.

**Table 5 antioxidants-12-00365-t005:** Total cost of the production and use of UGE per sample and the different dry extract yield per liter of liquid extract.

	Yield	
	20 g	30 g
	EUR/L	
Sample A	0.47	0.31
Sample B	0.94	0.63
	EUR/bottle (0.75 L)	
Sample A	0.36	0.24
Sample B	0.71	0.47

## Data Availability

Data is contained within the article.

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
