# Peer review of "Replacement of SO2 with an Unripe Grape Extract and Chitosan during Oak Aging: Case Study of a Sangiovese Wine"

_antioxidants, 2023, doi:10.3390/antiox12020365_

Round 1

Reviewer 1 Report

This study focuses on the use of an unripe grape extract as a stabilizing additive during the aging of a Sangiovese wine in comparison with sulfur dioxide.

In my opinion, this manuscript should undergo some important revisions to be accepted for publication.

Abstract: “… and to the fact that they are not harmful to personal health.” Do the authors mean human health?

Line 52-53: “These compounds are extensively used due to their efficiency, mainly when they are added to the wine together sulfur dioxide.” There is a word missing in the end of the sentence.

Line 116-119: In my opinion, the wine chemical parameters should be presented in a Table at the Results and Discussion section.

Line 166: Please replace “Four mL” by “4 mL”.

Line 167: “A volume of 1 mL of diluted FC reagent…” If I understood properly, the authors are referring to the Folin-Ciocalteu reagent. However, at line 158, the authors define FC as flavanol cofactor content.

Line 222-223: Please replace “32,1 ± 7,8 years” by “32.1 ± 7.8 years”.

Tables: Please note that the results are not presented properly. The error or standard deviation should be presented with only 1 significant number, and the result in accordance. For instance: 73.5 ± 0.38 should be 73.5 ± 0.4; and 18.03 ± 1.19 should be 18 ± 1.

3.43±0.00 is not acceptable. The error or standard deviation is not shown.

Line 435: Please correct “On the contrary, in the A sample containing UGE and chitosan, B. bruxellensis was not detected.”

Table 4: p-values should be presented with “.” instead of “,”, correct?

Table 5: Please replace “l” for “L”.

Author Response

Reviewer 1

This study focuses on the use of an unripe grape extract as a stabilizing additive during the aging of a Sangiovese wine in comparison with sulfur dioxide.

In my opinion, this manuscript should undergo some important revisions to be accepted for publication.

Abstract: “… and to the fact that they are not harmful to personal health.” Do the authors mean human health?

Thank you. Yes, the authors mean human health. We have changed the sentence in text according to your comment (see line 24).

 Line 52-53: “These compounds are extensively used due to their efficiency, mainly when they are added to the wine together sulfur dioxide.” There is a word missing in the end of the sentence.

The authors added the missing word (with) in the text (see lines 54).

Line 116-119: In my opinion, the wine chemical parameters should be presented in a Table at the Results and Discussion section.

Thank you for your comment. The authors have performed the analyses on a single wine sample, consequently the effect of the different antioxidants cannot be correlated with the chemical characteristics of the wine. Therefore, the authors think that these data are more appropriate to be shown in the Material and Methods section than in the Results and Discussion section.

Line 166: Please replace “Four mL” by “4 mL”.

The authors replaced “four mL” by “4 mL” and modified the sentence as follows: “A volume of 4 mL” (see line 168).

Line 167: “A volume of 1 mL of diluted FC reagent…” If I understood properly, the authors are referring to the Folin-Ciocalteu reagent. However, at line 158, the authors define FC as flavanol cofactor content.

 Thank you. The authors replaced “FC” by “Folin-Ciocalteu” in the text (see line 170).

Line 222-223: Please replace “32,1 ± 7,8 years” by “32.1 ± 7.8 years”.

 Thank you.  The authors have made the corrections (see lines 221-222).

Tables: Please note that the results are not presented properly. The error or standard deviation should be presented with only 1 significant number, and the result in accordance. For instance: 73.5 ± 0.38 should be 73.5 ± 0.4; and 18.03 ± 1.19 should be 18 ± 1.

3.43±0.00 is not acceptable. The error or standard deviation is not shown.

The authors thank the reviewer for these comments. However, the authors preferred to show the data following the usual rules of instrumental analysis (Cozzi, R., Protti, P., Ruaro, T. Analisi chimica strumentale. Zanichelli Ed. 2007). According to these rules, the significant numbers of a measurement are the certain numbers and the first uncertain number. These numbers depend on the sensitivity of the measurement method, that is, the smallest amount of a substance that can be accurately measured by that method.  Given the sensitivity of the methods used in this study, the authors decided to show the standard deviation with two decimal numbers, and the results in accordance. Furthermore, in the case of 3.43±0.00, the authors presented a very small standard deviation, that is lower than 0.005. Similar standard deviations are frequently presented in papers by other authors that you can find also cited in our article.

Line 435: Please correct “On the contrary, in the A sample containing UGE and chitosan, B. bruxellensis was not detected.”

The sentence was corrected as follows: “On the contrary, in the A sample containing UGE and chitosan, as well as in the TQ sample containing sulfur dioxide, B. bruxellensis was not detected” (see lines 444-446).

Table 4: p-values should be presented with “.” instead of “,”, correct?

Thank you. The authors presented the p-values with “.” Instead of “,”.

 Table 5: Please replace “l” for “L”.

Thank you.  The authors replaced “l” for “L”.

Reviewer 2 Report

The manuscript deals with the use of Unripe grapes extracts (UGE) to reduce SO2 use during wine barrel ageing. Authors tested for that the chemical, microbiological and sensory properties of the two treated wines (A 200mg/L UGE + 100 mg/L of chistosan ; B : 400 mg/L UGE): compared to a control sulfited wines (TQ). Results show there is poor modification among wines after 3, 6 and 12 months among the three typologies of wines and open an interesting way of using UGE as an alternative of SO2.

However, in its state, the manuscript could not be published and  required substantial modifications :

-       Explain the use of the chosen dose for UGE ? indicate minimum composition of UGE in terms of polyphenols, glutathione, vitamins and organic acids ? As the objective is to protect wine : explain how the antioxydant properties of wines measured with DPPH/FRAP and ABTS methods indicate this level of protection. Additionally could author comment the antioxydant properties of UGE in terms of ripening : what is its evolution? 

-       Replacing SO2 by UGE could be promising. What about the vintage effect on the stabilizing properties of  UGE? Do we need to investigate it in future or not?

-       As the objective is is to replace SO2 during barrel ageing, there is missing information on barrel (oak ? toasted ? new? ….). What about its variability in the results of the study. The barrel ageing step is not clear : are 3/6/12 months of barrel ageing or bottle ageing? Please precise.

-       The use or unripe grapes (UG) to contribute to circular economy of wine is interesting but should be removed or better explained with the same objectives of comparing to SO2. How the prices were calculated should be explained (L.484) and compared to SO2 use.

-       In introduction, the term “residues” is awkward. Better to use byproducts. (L59 61)

-       On results :

1)    Table 1, precise if the standard deviation represent a triplicate on biological or technical replicate?

2)    Figure 1 : please indicate units for ordinates of the figures, and perform the same statistical treatment as done in Table 1 to indicate if one period of ageing is significant for one of the four parameters analyzed among the three wines. What about the differences between each period of ageing of the antioxydant properties of each wines ?

3)    Figure 2 is strange to discuss later and not while discussing Figure 1.

4)    L.428 : precise how the 36 CFU/mL(L.429) or the 125 and 25 CFU/mL (L.433) of Brettanomyces bruxellensis yeasts were analyzed?

5)    L.484 : precise how the price of  € 0.79 to € 1.18 per gram of dried antioxidant extract (L.484) was calculated?

Author Response

Reviewer 2

The manuscript deals with the use of Unripe grapes extracts (UGE) to reduce SO2 use during wine barrel ageing. Authors tested for that the chemical, microbiological and sensory properties of the two treated wines (A 200mg/L UGE + 100 mg/L of chistosan ; B : 400 mg/L UGE): compared to a control sulfited wines (TQ). Results show there is poor modification among wines after 3, 6 and 12 months among the three typologies of wines and open an interesting way of using UGE as an alternative of SO2.

However, in its state, the manuscript could not be published and  required substantial modifications :

-       Explain the use of the chosen dose for UGE ? indicate minimum composition of UGE in terms of polyphenols, glutathione, vitamins and organic acids ? As the objective is to protect wine : explain how the antioxydant properties of wines measured with DPPH/FRAP and ABTS methods indicate this level of protection. Additionally could author comment the antioxydant properties of UGE in terms of ripening : what is its evolution? 

The authors thank the reviewer for the comments. The explanation about the use of the chosen dose for UGE  is already reported at lines 285-288.

The authors reported in Materials and Methods, Results and Discussion, and References sections five papers where the composition of UGE is already published. For this reason, the authors decided to add in this paper only the antioxidant activity of UGE, as follows: “The phenolic concentration of UGE was 20.4 mg CAT eq/g of powder and the antioxidant activity, evaluated by DPPH method, was 33.8 mmol Trolox/g.”. (see lines 282-283).

The authors added a comment to explain how the antioxidant properties of wine indicate the level of protection as follows: “The antioxidant properties of wines, measured by the DPPH, FRAP and ABTS methods, during the aging period (Figure 1) indicated that the addition of UGE (400 mg/L) or UGE (200 mg/L) and chitosan (100 mg/L) can give a level of protection from oxidation similar to that exerted by SO2. It’s well documented that the antioxidant acitivity of phenol compounds can be related to different properties including the capacity of scavenging superoxide radicals, consuming dissolved oxygen, reducing  ferric iron (Fe3+) to ferrous iron (Fe2+) and chelating Fe2+ (6).” (see lines 389-395).

-       Replacing SO2 by UGE could be promising. What about the vintage effect on the stabilizing properties of  UGE? Do we need to investigate it in future or not?

The authors thank the reviewer for this comment and surely it should be necessary to investigate it in the future The results of the work are still limited to the aging phase of a Sangiovese wine. The authors added a sentence in the Conclusion section: “In the future, it should be investigated the evolution of wine aged with the addition of UGE in substitution of SO2 during vintage.” (see lines 539-541).

-       As the objective is is to replace SO2 during barrel ageing, there is missing information on barrel (oak ? toasted ? new? ….). What about its variability in the results of the study. The barrel ageing step is not clear : are 3/6/12 months of barrel ageing or bottle ageing? Please precise.

Thank you for the comment. The authors added information about the barriques used: “All the barriques (Tonnellerie Baron, Les Gondes, France) used for the study were made with French oak. The barriques were two years old and medium toasted.” (see lines 122-124). For this reason, the oak characteristics did not introduce variability in the results of our study.

The authors added the information about aging as follows: “Chemical, microbiological and sensory analyses were performed at the start (T0), and after 3 (T3), 6 (T6) and 12 (T12) months of aging in barrique.” (see lines 129-130). 

-       The use or unripe grapes (UG) to contribute to circular economy of wine is interesting but should be removed or better explained with the same objectives of comparing to SO2. How the prices were calculated should be explained (L.484) and compared to SO2 use.

The authors thank the reviewer for the comment. At lines 80-82 it is already explained how the UGE contributes to the circular economy, reducing input and waste of the oenological process.

Please, about the prices see the answer below: (5), L.484).

-       In introduction, the term “residues” is awkward. Better to use byproducts. (L59 61)

The authors replaced the term “residues” with “byproducts” (see lines 61, 63, 85-86).

-       On results :

1)    Table 1, precise if the standard deviation represent a triplicate on biological or technical replicate?

The authors thank the reviewer for the comment. Standard deviation was calculated on all the data obtained.  Measurements (technical) were made on three different samples picked up at the different stages of aging for each barriques (TQ, A and B), which were set up in duplicate (biological).

2)    Figure 1 : please indicate units for ordinates of the figures, and perform the same statistical treatment as done in Table 1 to indicate if one period of ageing is significant for one of the four parameters analyzed among the three wines. What about the differences between each period of ageing of the antioxydant properties of each wines ?

Thank you for your comment. The authors have added units to the figures and performed the statistical treatment which are now showed in Figure 1. The authors added a sentence that explain the very few differences between each period of aging: “The evolution of total phenols and antioxidant activity during aging is showed in Figure 1. Over the course of twelve months very few significant differences in the antioxidant activity of samples were detected. In particular, it is possible to note that after three months, samples A and B had a significantly lower concentration of total phenols in comparison to that of sample TQ. Moreover, at the start (T0) and after three months, the values of ABTS were significantly lower in sample B.” (see lines 366-371).

Furthermore, the authors added sentences to describe differences between the antioxidant activity of the samples for each period of aging, as follows: “The evolution of TP (Figure 1a) of all the wine samples showed a positive trend, mainly from the third month. In general, the TP content of the wines increased over aging due to contact with toasting oak [44]. The antioxidant activity evaluated by DPPH, FRAP and ABTS (Figure 1b, c, and d) in general decreased from the start to the end of the aging period. The highest decreasing was observed in the antioxidant activity assayed by DPPH method.” (see lines 384-389).

The authors performed statistical treatment of data showed in Figure2 and modified this Figure by adding this information.

3)    Figure 2 is strange to discuss later and not while discussing Figure 1.

The authors thank the reviewer for the comment. The authors have moved Figure 1 near to Figure 2, together with its results and comments. The author have added a description and discussion of the results showed in Figure 2, as follows: “The evolution of some parameters of the wine treated with different stabilizing products during the aging period is shown in Figure 2. The sample TQ had significantly lowest CI and highest L* values at all stage of aging (Figure 2a and c). These results were a consequence consistent with the further addition of SO2 made at the fourth month of aging, and its effect of bleaching on anthocyanins [42, 43]. The differences in the AL, COP and PP content among the samples were detectable as early as the third month and they were maintained until the end of aging (Figure 2). The TQ samples had the significantly highest AL and lowest PP values from T3 to T12 (Figure 1f and h). Moreover, COP values of sample TQ were significantly highest from the third month to the end of aging (Figure 1g). During this period, the rate of stable polymeric pigments (PP) formation in the samples A and B was higher than that of TQ, and was accompanied by a decrease of free anthocyanins (AL). These results highlighted that the addition of UGE could contribute to a faster stabilization of color. Hence, at the same stage of evolution, the wine aging with the addition of UGE contained more stable pigments towards oxidation phenomena in comparison to those of wine sample (TQ) aging with SO2 [31, 47].” (see lines 396-410).

4)    L.428 : precise how the 36 CFU/mL(L.429) or the 125 and 25 CFU/mL (L.433) of Brettanomyces bruxellensis yeasts were analyzed?

Thank you for your comment. The authors have added the method used to identify B. bruxellensis in the Materials and Methods section, as follows: “To enumerate B. bruxellensis, a representative number of colonies showing green colour, round morphology, and yellow halo, were picked up and analysed by RFLP-PCR of rITS, for their identification [39] (see lines 216-218).”

5)    L.484 : precise how the price of  € 0.79 to € 1.18 per gram of dried antioxidant extract (L.484) was calculated?

We thank the reviewer for the suggestion to mention here how the costs were quantified. To this end, the sentence has been modified by adding some details and showing how the calculation was done according to the methodology cited in section 2.14. The authors added the following sentence: “The full cost analysis carried out for the case study examined in this paper revealed that the farm, in order to produce the necessary UGE by itself, has to incur a total cost ranging from 0.79 to 1.18 per gram of obtained dried antioxidant extract. As explained in the paragraph 2.14, this cost is calculated by examining the entire process of producing the extract, starting from harvesting the unripe grapes to obtaining the final product: process inputs (capital and labor) were considered for each operation and the direct and indirect costs to be incurred by the entrepreneur for their use were quantified. The total cost range is mainly due to the dry extract yield, which can range from 20 to 30 grams per liter of processed liquid extract.” (see lines 492-500).

Round 2

Reviewer 1 Report

According to the authors own words: “... the significant numbers of a measurement are the certain numbers and the first uncertain number.” In the case of 3.43±0.00 it supports what I have questioned.

The authors may reply that “0” is a number, but then I can ask: Why not present the result as 3±0?

If the standard deviation is lower than 0.005 that is an excellent result. The authors should show it properly as, for instance, 3.430±0.004

Author Response

Dear Editor and reviewer,

we provide details of the changes made for your approval:

Tables: the authors modified the results showed in Tables 1 and 2 according to the reviewer’s comment.

Best regards,

Giovanna Fia

Reviewer 2 Report

Thanks to authors for the precisions brought to the questions. It was not so easy in the reviewing process to track the changes done on the revised manuscript. The last sentence in conclusion is awkward. Better to say : "The substitution of SO2 by UGE, should be carried in future studies taking in consideration the vintage effect in wines"

Author Response

Dear Editor and reviewer,

we provide details of the changes made for your approval.

Lines 539-541. The authors replaced the sentence as follows: “The substitution of SO2 by UGE, should be carried in future studies taking in consideration the vintage effect in wines.”

Best regards,

Giovanna Fia